# Activity of Cytosolic Ascorbate Peroxidase (APX) from *Panicum virgatum* against Ascorbate and Phenylpropanoids

**DOI:** 10.3390/ijms24021778

**Published:** 2023-01-16

**Authors:** Bixia Zhang, Jacob A. Lewis, Frank Kovacs, Scott E. Sattler, Gautam Sarath, ChulHee Kang

**Affiliations:** 1Department of Chemistry, Washington State University, Pullman, WA 99164, USA; 2Chemistry Department, University of Nebraska-Kearney, Kearney, NE 68849, USA; 3Wheat, Sorghum and Forage Research Unit, U.S. Department of Agriculture—Agricultural Research Service, Lincoln, NE 68583, USA

**Keywords:** ascorbate, switchgrass, ROS, stress, phenylpropanoids, ascorbate peroxidase, lignification, X-ray crystallography, oxidation

## Abstract

APX is a key antioxidant enzyme in higher plants, scavenging H_2_O_2_ with ascorbate in several cellular compartments. Here, we report the crystal structures of cytosolic ascorbate peroxidase from switchgrass (*Panicum virgatum* L., *Pvi*), a strategic feedstock plant with several end uses. The overall structure of PviAPX was similar to the structures of other APX family members, with a bound ascorbate molecule at the ɣ-heme edge pocket as in other APXs. Our results indicated that the H_2_O_2_-dependent oxidation of ascorbate displayed positive cooperativity. Significantly, our study suggested that PviAPX can oxidize a broad range of phenylpropanoids with δ-meso site in a rather similar efficiency, which reflects its role in the fortification of cell walls in response to insect feeding. Based on detailed structural and kinetic analyses and molecular docking, as well as that of closely related APX enzymes, the critical residues in each substrate-binding site of PviAPX are proposed. Taken together, these observations shed new light on the function and catalysis of PviAPX, and potentially benefit efforts improve plant health and biomass quality in bioenergy and forage crops.

## 1. Introduction

The production of reactive oxygen species (ROS), such as the superoxide radical (O_2_^−^), the hydroxyl radical (OH) and hydrogen peroxide (H_2_O_2_) as by-products of various enzymatic reactions in cellular compartments, or produced as part of a stress/defense response is one the most critical homeostatic equilibria in plant cells [1]. Elevated levels of H_2_O_2_ in cellular compartments is especially dangerous, as it is the sole ROS that can diffuse through biological membranes, increase in various subcellular compartments [2] and disturb several critical cellular processes, including photosynthesis [3,4]. Ascorbate is a well-known antioxidant and radical scavenger. It can react spontaneously with superoxide and other radicals. Significantly, the rate constant for ascorbate reacting with H_2_O_2_ is low under physiological conditions (2 M^−1^s^−1^) [5]. Thus, ascorbate acts as an electron donor in the degradation of H_2_O_2_ via the dynamic interactions of catalase and ascorbate peroxidase (APX). In plants, APXs are widely present in the cytosol, peroxisome, mitochondria and chloroplast, where they play a critical role in maintaining cellular redox homeostasis by removing H_2_O_2_ produced by metabolic processes such as those occurring in chloroplasts [2,6]. Although various substrates are utilized as preferred substrates, the H_2_O_2_-dependent catalytic reactions of peroxidases follow three common steps: (1) the enzyme is oxidized by H_2_O_2_ and forms intermediate Compound Ⅰ; (2) Compound Ⅰ is reduced by a substrate forming Compound Ⅱ, resulting in one-electron oxidized substrate; (3) Compound Ⅱ is then reduced by a second substrate, forming a water molecule and returning the enzyme to a resting state as ferric (Fe(III)) heme iron. Compound I in Class I APXs exists as a ferryl (Fe((IV)) heme and a porphyrin π-cation radical, and its Compound II contains only a ferryl oxo species [7,8]. In APX, the ascorbate molecule binds to a location referred to as the γ-heme edge [9], and electron transfer from ascorbate to the heme in Compounds I/II in APX is presumably via the heme propionates.

It is generally accepted that APX produces two monodehydroascorbate (MDHA) radials per one H_2_O_2_, conducted through two successive radical transfers from the Compound I intermediate oxidized by one H_2_O_2_ molecule [10]. However, sorghum APX (SbAPX) can produce a C2-hydrated bicyclic hemiketal form of dehydroascorbic acid, indicating two successive electron transfers from a single bound ascorbate [11]. In aqueous solution, MDHA radicals are readily converted to dehydroascorbic acid (DHA1) and reduced to ascorbate by DHA reductase or nonenzymatically by glutathione (GSH) in both plant and animal tissues [12]. Dehydroascorbic acid (DHA1) is hydrated rapidly at the 2′-position, due to its high nucleophilicity in aqueous solutions, to form the hydrated form DHA2, which in turn forms a bicyclic hemiketal under physiological conditions [13]. Although little information is available about the physiological role and predominant form of DHA in vivo, it has been proposed that a balanced ratio between ascorbate and DHA is as critical as NAD(P)H/NAD(P) and GSH/GS-SG [13,14]. It is known that APXs can catalyze the oxidation of various aromatic species, including pyrogallol, guaiacol, *p*-cresol, *o*-dianisidine, and 2,2′-azinobis (3-ethylbenzothiazoline-6-sulfonic acid) (ABTS), and several non-physiological substrates [15,16]. In addition, APX was recently identified as displaying 4-coumarate-3-hydroxylase (C3H) activity in *Arabidopsis thaliana* and *Brachypodium dystachion* [17]. This finding implied the existence of a shunt in the lignin pathway to bypass multiple enzymes to form caffeic acid. A recent report on *P. tomentosa* mitochonrial APX (PtomtAPX) showed that the mitochondrial APX could relocate to the cell wall to assist lignin polymerization in secondary cell wall formation and xylem development [18]. As SbAPX, PtomtAPX catalyzes the dimerization of monolignols such as *p*-coumaryl alcohol, coniferyl alcohol and sinapyl alcohol, utilizing H_2_O_2_ as a substrate [11,18]. Thus, PtomtAPX displays a dual functionality, linking the scavenging of the ROS with lignification of the cell wall. General reaction mechanism of APX are as follows:
APX (Fe^III^ Porphyrin) + H_2_O_2_ → APX (Fe^IV^=O Porphyrin^•+^) + H_2_O(1)
APX (Fe^IV^=O Porphyrin^•+^) +HS → APX (Fe^IV^=O Porphyrin) + S^•^(2)
APX (Fe^IV^=O Porphyrin) + HS → APX (Fe^III^ Porphyrin) + S^•^ + H_2_O(3)

In Reaction (1), the enzyme is oxidized by H_2_O_2_ to form the intermediate oxidized state where ^•+^ represents a porphyrin phi-cation radical and Fe^IV^ = O represents a ferryl oxo heme. In Reaction (2), the porphyrin is reduced by a general substrate to form a one-electron oxidized state. In Reaction (3), the reaction center is reduced again by another general substrate, forming a water molecule and the resting ferric state.

As a perennial grass adapted to many regions in North America, switchgrass (*Panicum virgatum* L.) has been targeted as a model grass for biofuels [19]. Switchgrass requires minimal agricultural input and thus can be sustainably grown on marginal croplands [20]. Continuous improvement of the yield and quality of switchgrass is a pressing need in order for the U.S to meet its national goal of replacing a portion of petroleum gasoline with biofuel [21]. One plausible approach for achieving this goal is to manipulate plant lignin biosynthesis, leading to changes in lignin content and its monomer composition, which could significantly improve biofuel yields [22]. Lignin is an aromatic polymer of plant cell walls that accounts for 20% to 30% of all terrestrial plant biomass [23,24,25]. Lignin is cross-linked to cell wall hemicellulose, resulting in both structural and protective fortification of plant cells [26]. However, lignin is also a major source of recalcitrance for the biochemical conversion of herbaceous biomass into biofuels [22]. Thus, optimizing lignin levels in biomass will help to mitigate the negative impact of lignin in the biochemical conversion process of lignocellulose to bioethanol or, alternatively, will help increase the energy content of herbaceous biomass [27]. Lignin deposition occurs in the apoplastic space, either as a normal process of secondary cell wall accretion, or in response to stress. Switchgrass plants can experience significant oxidative stress when infested with insects [28] or microbes [29], and it is likely that PviAPX1 plays a central role in mitigating oxidative stress. Considerable resources have been devoted to improving switchgrass biomass yields and quality [30,31,32,33]. Although vegetative biomass was targeted for biochemical conversion to ethanol [34], switchgrass biomass is suitable for producing butanol [35], or products derived by pyrolysis [36].

Here, we report comprehensive characterization of APX from *Panicum virgatum* (PviAPX) through determining its crystal structure and examining enzyme kinetics toward ascorbate and phenylpropanoids.

## 2. Results

### 2.1. Enzyme Preparation and Spectral Properties

The original PviAPX clone [37] was derived from an EST collection [38] and annotated as *Pavirv00022559m.g.* in the first annotation of the *Panicum virgatum* genome. However, these tags have since been removed from databases, and the *PviAPX* gene corresponds to *Pavir.9KG480900* in the latest genome annotation (version 5.1, Phytozome.org). A recombinant PviAPX enzyme was purified from *Escherichia coli* cells containing an expression vector harboring *PviAPX* with an N-terminal 6×His tag encoded at its 5′ end. Purified PviAPX displayed a Soret band with a maximum at 404 nm, with Q bands at 507 nm and 540 nm (Figure 1), which indicated its heme in the ferric (Fe(III)) state. There was no obvious change in the spectrum when ascorbate was added to the protein.

### 2.2. Overall Structure and Heme Environment of Apo-Form PviAPX

Numerous screenings for initial crystallization of PviAPX did not yield any hit. Thus, to assist crystallization, site-directed mutagenesis was conducted on the specific residues that are common in the existing APX crystal structures, but not seen in PviAPX. Three mutations, C4S, C168A and K229D were selected, which are far from both the heme-binding and ligand-binding pockets. In addition, the construct contained two conservative mutations, E14D and V221A, that were part of the cloned recombinant PviAPX [37]. Two corresponding amino acids, Glu-14 and Val-221, in the 16 other full-length APXs found in the switchgrass genome, are variable. As expected, the corresponding mutant PviAPX displayed similar redox-related spectral properties to those of the wild-type (Figure 1). It was crystallized in a P1 space group with six molecules in the asymmetric unit, and its three-dimensional structure was determined at 2.5 Å resolution (PDBID: 8FF6) (Table 1) (Figure 2A). All the amino acid residues of the sequence from N-terminus Met to C-terminus were verified from the electron density map. The average root mean square deviation (rmsd) value among six PviAPX molecules was 0.36 Å. Analysis of the molecular interfaces of PviAPX in crystal lattices with a PDBePISA server [39], which evaluates interactions between neighboring monomers in the crystal lattices for the purpose of predicting biologically relevant oligomeric states, was conducted. No significant complex formation among PviAPX is suggested, and thus PviAPX likely exists as a monomer in solution (solvation free energy gain = −5.1 kcal mol^−1^ and interface complexation significance score = 0), which is consistent with a previous report [37].

The overall fold of PviAPX contained eleven α-helices (αA-αI), three 3_10_ helices (ηA-ηC) and two short antiparallel β-strands (βA and βB) (Figure 3). Its catalytic center possessed a penta-coordinated Fe(III) ligated to four nitrogen atoms of the porphyrin and the imidazole sidechain of the proximal His-163, which was similar to those observed among other APXs. The rmsd values of PviAPX for SbAPX (PDBID: 8DJR), rsAPX (PDBID: 1OAG), rpAPX (PDBID: 1APX) and tbAPX (1IYN) are 0.37 Å, 0.64 Å, 0.64 Å, and 0.69 Å, respectively. As in other structurally characterized peroxidases, His-163 from ηC, Trp-179 from the loop between βB and αG, and Asp-208 from αH in the proximal heme pocket of PviAPX were present at the same position with similar sidechain orientations. On the distal side, Arg-38, Trp-41, and His-42 from αB are also conserved (Figure 3). The corresponding residues of two short β-strands, which dicot APXs have adjacent to the 7-propionate of the heme, were conserved as ^167^Ala-Ala^168^ and ^177^Gly-Pro^178^ in dicot APX, but not among monocot APX (sorghum and switchgrass) (Figure 3). In addition, a high level of heterogeneity in the amino acid sequence was observed in helices and loops located distal from heme.

The electron density of the apo-form PviAPX revealed a metal ion from the early stages of the refinement which was located at the distal side and positioned 13 Å from heme iron (Figure A1). An Na^+^ ion was placed, since this was the only cation present in our final purification step and crystallization procedures. The Na^+^ ion was coordinated by the sidechains of Thr-164, Thr-180, Asn-182, Val-185 and Asp-187, all of which are conserved among APXs. This position of the metal ion is very close to the proximal Ca^2+^ ion observed in many Class III peroxidases [42,43,44,45].

Consistent with the spectral signals of the purified mutant and wild-type PviAPX, the apo-form crystal structure showed what was presumably the resting ferric state (Fe(III)). A bound but non-coordinating water molecule was positioned distal to the heme at a distance of 2.2 Å from the iron, which was similar to the observed ferric heme species in rsAPX [46,47]. The N^e^ atom of His-42 on the distal side was hydrogen bonded to a water molecule, which in turn hydrogen bonded to the heme-bound water molecule. In addition, this iron-bound oxygen atom of the water was 3.34 Å and 3.04 Å from the N^e^ atom of His-42 and Trp-41, respectively. The hydrogen-bond network was further consolidated with three ordered water molecules found in distal portion of the pocket, which was previously reported in the rsAPX structure [9].

### 2.3. Ascorbate Complex Structure of PviAPX

The ascorbate binary complex structure of PviAPX was obtained by soaking the above-mentioned apo crystal in the crystallization mother liquor containing 1 mM ascorbate solution. Its three-dimensional structure was determined at 2.2 Å resolution (PDBID: 8FF7) (Figure 2B). Electron density corresponding to a bound ascorbate was identified at the ɣ-heme edge position. This pocket was surrounded by Cys-32, Pro-34, Leu-35, His-169 and Arg-172, all of which are highly conserved among the compared APX structures (Figure 3). 2-hydroxyl groups of the ascorbate furan ring were hydrogen-bonded to the 6-propionate group of the heme and the sidechain of His-169 and Arg-172. In addition, the carbonyl oxygen of the furan ring of ascorbate was within hydrogen-bond distance from the backbone nitrogen of Leu-35. The 6-hydroxyl group of the ascorbate was hydrogen-bonded to the backbone of Lys-30, which is similarly observed in rsAPX (PDBID: 1OAF) [9,48]. In addition, four water molecules nearby established hydrogen bonds linking the ascorbate to PviAPX enzyme (Figure 4). Compared to the apo-form PviAPX, association of this ascorbate molecule displaced two water molecules without any noticeable shift of side chains surrounding the pocket; this is different from the structure of rsAPX, wherein the side chain of Lys-30 swung to establish an extra hydrogen bond with bound ascorbate.

### 2.4. Steady-State Kinetics of H_2_O_2_/Ascorbate

Steady-state kinetics for the H_2_O_2_-dependent oxidation reaction of ascorbate were performed for both the wild-type and mutant PviAPX proteins (Figure 5). The profile indicated an allosteric sigmoidal kinetic model instead of a Michaelis–Menten model [49]. When the concentration of ascorbate was increased to 1200 μM, a substrate inhibition was observed in both the wild-type and mutant PviAPX. The substrate inhibition was more evident in the wild-type PviAPX. However, the K_i_ cannot be determined due to a rapid decrease in activity. Thus, the fitting was conducted only with the ascorbate concentration ranging from 0 to 1200 μM (Figure 5). The fitting of wild-type from the Hill plot resulted in a V_max_ of 12110 ± 3803 min^−1^, a Hill slope of 1.129 ± 0.177, and K_half_ of 1049 ± 573 μM. Fitting of the mutant data from the Hill plot resulted in a V_max_ of 7239 ± 1997 min^−1^, a Hill slope of 1.375 ± 0.421, and K_half_ of 508.0 ± 243.5 μM. A Hill slope of greater than 1 indicated positive cooperativity of multiple binding sites [11,50].

### 2.5. The Oxidation Reaction of PviAPX for Phenylpropanoids

The addition of PviAPX and H_2_O_2_ to coniferyl alcohol containing phosphate buffer (pH 7.5) instantly changed the color of the reaction and yielded product peaks together with a substantial reduction of substrate peak in the HPLC profile. Therefore, other phenylpropanoid compounds, such as *p*-coumaric acid, caffeic acid, ferulic acid, sinapic acid and coniferyl aldehyde, were tested in the same buffer solution containing PviAPX and H_2_O_2_. The result showed that PviAPX was able to utilize the tested phenylpropanoids as substrates. Among those phenylpropanoids, ferulic acid, sinapic acid and coniferyl alcohol had the highest activity (Figure 6). LC-MS analysis indicated the molecular weight of the products was twice the substrate molecular weight (Figure A2), which indicated these products were formed through radical dimerization.

To ensure the validity of the mutant PviAPX structure represented the wild-type features, oxidation activity assay with the same phenylpropanoid compounds was conducted for the mutant and wild = type PviAPX. The result showed that the mutant selected for crystallization had similar activity as the wild-type in all reactions with different substrates (Figure 6).

### 2.6. Molecular Docking for Phenylpropanoids

To further investigate the binding preference of phenylpropanoids, molecular docking was performed with the structural coordinates of the apo-form PviAPX. All the phenylpropanoid intermediates of the monolignol pathway were docked to investigate the affinity of acids, alcohols and aldehydes (Table 2). The preferred docked position of *p*-coumaric acid, caffeic acid, ferulic acid and sinapic acid was the δ-meso site, not the ɣ-heme edge position of PviAPX (Figure 7). Their phenolic rings faced toward the heme and the phenol oxygen of those acids established a hydrogen bond with Arg-38; their propenyl oxygen also established a hydrogen bond from with His-42. The binding energy of *p*-coumaric acid, caffeic acid, ferulic acid and sinapic acid was −5.5, −5.8, −5.7 and −5.8 kcal/mol, respectively. All the alcohols and aldehydes substrates tested, except for caffeoyl aldehyde, showed the most negative free energy when their phenolic rings faced the heme group. In the case of caffeoyl aldehyde, the phenolic ring facing the heme was the next most favorable orientation (Table 2). When their phenolic rings were facing the heme group, these compounds all showed the same binding pattern as the aforementioned acids, interacting with both Arg-38 and His-42. Trp-41 could also form indirect hydrogen bonds with the substrates through a water molecule (Figure 7M). In general, the aldehydes were less preferred relative to acids or alcohols, and an increasing degree of substitution on the phenolic ring decreased the binding energy based on this analysis.

## 3. Discussion

### 3.1. Potential Physiological Roles of Cytosolic APX

Cytosolic APXs (cAPX) have been reported to have diverse metabolic functions and potential regulation by nutrient status and post-translational modifications (PTM) in plants. Recently, it has been demonstrated that cAPX is central to the monitoring of ROS levels in cells, especially when under pathogen infection, indicating a key role in defense responses [51]. Overexpression of cAPXs improves stress responses via ROS detoxification [52,53]. The *Oncidium* cAPX can utilize both ascorbate and glutathione as substrates, thereby improving abiotic stress tolerance when overexpressed in Arabidopsis. Other researchers have demonstrated that cAPX can function as a chaperone, in addition to its documented enzymatic activities, and that enzymatic activity but not chaperone function was affected by PTMs [54]. The nitrogen status of the plant also intersects with cAPX, with low nitrogen levels contributing to changed ascorbate levels, primarily through expression levels of cAPX [55]. The ability of cAPX to utilize phenolic compounds also points to a role in cell wall lignification [11,17]. Overexpression of a *Rheum australe* cAPX and a superoxide dismutase from *Potentilla atrosanguinea* in Arabidopsis enhanced ectopic lignification as compared to WT Arabidopsis plants [52]. These data indicate the role of cAPXs in diverse aspects of plant growth. PviAPX is induced during seed germination in response to applied H_2_O_2_ [56]. *PviAPX* expression is induced during leaf senescence [57], and biotic stress [28,58]. Since biotic stress in switchgrass also activates other genes associated with phenylpropanoid metabolism and increases levels of defensive compounds synthesized from caffeic acid [28,58,59], it is plausible that PviAPX is involved in these processes along with its expected role in ROS homeostasis.

### 3.2. Two Independent Binding Sites of PviAPX

As shown in Figure 4, the crystal structures of SbAPX displayed that the supplemented ascorbate was associated first with the ɣ-heme edge site of the apo-form PviAPX, replacing two water molecules. 2- and 3-hydroxyl groups of the bound ascorbate furan ring were hydrogen bonded to the 6-propionate group of the heme and the sidechain of Arg-172. This ɣ- site is also used in the manganese peroxidase/Mn^2+^complex, where the Mn^2+^ was both coordinated with 6-propionate and nearby carboxylate oxygens of sidechains [60].

The oxidation reaction mechanism of the ascorbate at this site has been proposed to follow the formation of a porphyrin π-cation radical in Compound I, and an electron is transferred from bound ascorbate via the propionyl group to the heme [46]. Our results indicated that the H_2_O_2_-dependent oxidation of ascorbate displayed positive cooperativity indicating more than one ascorbate binding site (Figure 5).

Both R172A and R172S mutations confirm their critical role in stabilization of the bound ascorbate [61]. However, the corresponding activity for other aromatic substrates are not affected, differentiating activity against ascorbate from other substrates. The R172S protein exhibited negligible ascorbate peroxidase activity but showed near wild-type activity toward other aromatic substrates. In addition, there is another potential binding site, the δ-meso site, which is part of a hydrogen bond network with the sidechains of Trp-41 and His-42 and water molecules on the distal side of the heme. This δ-meso site is the reported binding site for the salicylhydroxamic acid in rsAPX (PDBID: 1V0H) [48] and the *Arthromyces ramosus* peroxidase (PDBID: 1CK6) [62], benzhydroxamic acid (BHA) in HRP [63], ferulic acid in HRP (PDBID: 6ATJ) [64], and the molecular docked phenylpropanoids in PvPRX [42]. As shown in Figure 7, our results also showed that the docked phenylpropanoids were positioned at the δ-meso position of PviAPX, where each molecule interacted with the heme iron indirectly via a water molecule. Therefore, those organic compounds at δ-meso can be radicalized directly or indirectly through the obvious proton-shuttle system established by nearby residues, including His-42, and water molecules can effectively abstract a proton from the ascorbate. Those interacting residues, Arg-38, Trp-41, Pro-132 and Asp-133, are completely conserved among PviAPX and the other APX enzymes we compared. The resulting radical products of phenylpropanoids were dimerized (Figure A2).

## 4. Materials and Methods

### 4.1. Recombinant Enzyme Expression and Purification

A PviAPX cDNA sequence was cloned into vector pET-30a(+) (MilliporeSigma, St. Louis, MO, USA) with N-terminal 6x-His tag for heterologous expression. The vector was introduced into *E. coli* Rosetta 2 (DE3) cells via transformation. Site-directed mutagenesis for R38L, W41F, H42A and R172A was performed by PCR using the primers designed by Agilent Quickchange Primer Design (https://www.agilent.com/store/primerDesignProgram.jsp accessed on 12 July 2022). A three-liter Luria-Bertani (LB) medium complemented with 25 μg mL^−1^ chloramphenicol and 50 μg mL^−1^ kanamycin was inoculated with 20 mL from an overnight culture. The cells were grown at 37 °C until the culture reached OD_600_ 0.8, and IPTG was added to a final concentration of 2 mM. After being induced for 4–5 h, the cells were harvested by centrifugation at 8000× *g* for 10 min at 4 °C. The cells were resuspended in buffer A (50 mM KP_i_, pH 8, 300 mM NaCl) and sonicated on ice for 30 min. (Model 450 sonicator; Branson Ultrasonics, Danbury, CT, USA) to release soluble protein. The cell debris was removed by centrifugation at 37,000× *g* for 1 h. The clear lysate was loaded on a nickel-NTA column (Qiagen, Germantown, MD, USA) and washed with three column volumes of Buffer A. The column was then washed by three column volumes of Buffer B (50 mM Kpi, pH 6, 300 mM NaCl, 10% (*v*/*v*) glycerol) followed by two column volumes Buffer B containing 20 mM imidazole. PviAPX was eluted with Buffer B containing 200 mM imidazole. During the purification, the His-tag was spontaneously cleaved, which could be due to a thrombin site between the His-tag and PviAPX. The protein was concentrated, buffer-exchanged against 5 mM potassium phosphate buffer pH 6.8 and loaded onto a hydroxyapatite column. PviAPX was eluted by a linear gradient of potassium phosphate pH 6.8, ranging in concentration from 5 to 25 mM, and the fractions with ratio OD_404_:OD_281_ > 2 were concentrated to 1 mL by using an Amicon 8050 ultrafiltration cell with a 10-kDa cutoff membrane (EMDMillipore, St. Louis, MO, USA). The concentrate was loaded onto a column containing Superdex™ 200 Increase 10/300 GL for further purification. During the purification, a 5-kD fragment was removed from the original recombinant APX. This truncated protein was loaded on a Ni-NTA column and was collected in the flow-through, which indicated that the truncation resulted in the loss of the N-terminal His-tag. The PviAPX without His-tag was buffer-exchanged against 20 mM Tris, pH 7.5, 50 mM NaCl (Buffer C).

### 4.2. Crystallization and Structure Determination

Prior to crystallization, the PviAPX was concentrated to 20 mg mL^−1^ using an Amicon 8050 ultrafiltration cell with a 10-kDa cutoff membrane (EMDMillipore, St. Louis, MO, USA). A commercial crystallization kit, Crystal screen HT (Hampton, Aliso Viejo, CA, USA), was used for crystal screening through the sitting-drop, vapor-diffusion method by Crystal Phoenix (Art Robbins Instruments, Sunnyvale, CA, USA). The initial crystal appeared in the condition B5 (0.2 M Li_2_SO_4_, 0.1 M Tris, pH 8.5 and 30% PEG 4000) at 4 °C. Then, the larger crystals were reproduced by the sitting-drop vapor-diffusion method with the same solution. Small crystals appeared in 2 days. The one-ascorbate structure was obtained by soaking apo crystals in cryoprotectant containing 1 mM ascorbate for 1 min, whereas the four-ascorbate structure was obtained by soaking apo crystal in cryoprotectant containing 1 mM ascorbate for 10 min. All the statistics of deposited structures are listed in Table 1.

### 4.3. Comparison of Activity among Different Phenylpropanoid Substrates of PviAPX

The H_2_O_2_-dependent reaction was conducted in 1 mL 50 mM potassium phosphate buffer, pH 7.5 containing 100 μM *p*-coumaric acid, caffeic acid, ferulic acid, coniferyl alcohol and coniferyl aldehyde, 1 mM H_2_O_2,_ and 0.2 μM of purified PviAPX and mutant. The reaction mixture was incubated at room temperature for 10 min and quenched by 30% (*v*/*v*) glacial acetic acid prior to injection to HPLC. The reaction mixture was injected onto a Luna^®^ 5 μm C18 reverse phase column and the product was monitored by HPLC (Hitachi Elite LaChrom L-2100; Hitachi High-Tech, Schaumburg, IL, USA) operating at a flow rate of 1 mL min^−1^, with a gradient of solvent A (0.1% trifluoroacetic acid in deionized water) and solvent B (100% acetonitrile) varying from 95% A and 5% B to 0% A and 100% B (*v*/*v*) over a period of 30 min. Substrates were quantified by detection at 320 nm using the Hitachi Elite LaChrom L-2400 detector (VWR, Atlanta, GA, USA). All experiments were performed in triplicate.

### 4.4. LC-MS for Determining the Products of H_2_O_2_-Dependent Oxidation of Phenylpropanoids by PviAPX

The H_2_O_2_-dependent reaction of PviAPX was conducted in 1 mL of 50 mM potassium phosphate buffer (pH 7.5) containing 1 mM H_2_O_2,_ and 0.2 μM of purified PviAPX (or mutant) with 100 μM of each *p*-coumaric acid, caffeic acid, ferulic acid, coniferyl alcohol and coniferyl aldehyde. The reaction mixture was incubated at room temperature for 10 min and quenched by 30 % (*v*/*v*) glacial acetic acid. Then, 100 μL of quenched reaction mixture was freeze-dried and redissolved in 0.1% formic acid.

The analysis of the reaction was done by LC-MS using Waters Xevo TQ-MS mass spectrometer interfaced with a Waters Acquity UPLC. An Ace Excel 1.7 SuperC18 (P/N EXL-1711-1003U, 100 mm × 3.0 mm i.d.) reverse phase HPLC column was used.

The mobile phase used was a binary gradient of water with 0.1% formic acid (Solvent A) and methanol with 0.1% formic acid (Solvent B). The solvent flow rate was 0.2 mL/min. and the initial solvent composition was 20%B/80%A (*v*/*v*) and was constant for the first minute following sample injection. Gradient elution was accomplished by increasing the ratio of solvent B to solvent A from 20:80 to 65:35 (*v*/*v*) over 10 min. using a linear gradient, and then increased to 100:0 over the next 4 min. The composition was maintained at 100% B for 5 min., returned to 80%A/20%B over the next 3 min. and held at that ratio for the next 3 min. The mass spectrometer was operated in positive ion mode with a capillary voltage of 3.2 and a cone voltage of 17 (unless specified differently in specific MS methods). The source temperature was 350 °C and API gas flow was 650 L/Hr. Single isotope reaction (SIR) methods were used for the analysis of the hypothesized dimerization products using the formula “(2 × monomer mass) + 1” to calculate mass of the ion to monitored.

### 4.5. Steady-State Kinetics of PviAPX

Kinetic assays of PviAPX with ascorbate were performed with 1 mM H_2_O_2_ in 50 mM potassium phosphate buffer, pH 7.0 at room temperature in a GENESYS™ 10S UV-Vis Spectrophotometer (Thermo Scientific, Waltham, MA, USA) spectrophotometer. The concentration of ascorbate varied from 10 to 1200 μM and the final concentration of PviAPX was 15 nM. The reaction was initiated by adding PviAPX. The activity was calculated by the rate of disappearance of ascorbate (ε = 2.8 mM^−1^ cm^−1^) at 290 nm during the first minute [65]. The corresponding activity for mutants was measured with the same setup above.

### 4.6. Molecular Docking of PviAPX

The phenylpropanoid intermediates of the monolignol pathway were docked into PviAPX by AutoDock Vina [66] for global search; ligands and grids were prepared for docking using AutoDock Tools [67]. The grid box was 100 Å × 100 Å × 100 Å. The exhaustiveness was set to 25 due to the large grid box.

## 5. Conclusions

This study comprehensively characterized the substrate-binding pockets and the plausible catalytic reaction mechanism of a switchgrass APX (PviAPX). Our data, with purified PviAPX indicating its ability to utilize both ascorbate and phenolics, also suggests that PviAPX could participate in different aspects of switchgrass metabolism. PviAPX not only dismutates H_2_O_2_ with ascorbic acid bound at ɣ-heme edge site, but it also can oxidize most phenylpropanoid intermediates in the monolignol pathway, such as *p*-coumaric acid, caffeic acid, ferulic acid, coniferyl alcohol and coniferyl aldehyde—which are bound at the δ-meso site—with rather similar efficiencies (Figure 7), Thus, APX, which is only known to be found in plants, algae and photosynthetic protists, can remove stress-generated H_2_O_2_ with ascorbates; however, it may also fortify cell walls with same H_2_O_2_ and phenylpropanoids in response to stress. The substitution of amino acids within the substrate-binding region of this dual function APX could potentially lead to changes in substrate affinity and result in the diversification of its function.

## Figures and Tables

**Figure 1 ijms-24-01778-f001:**
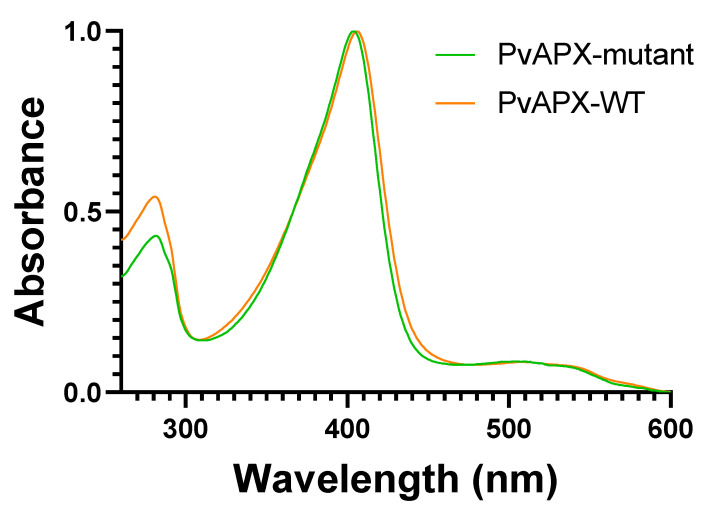
Spectrum of purified wild-type and mutant PviAPXs. The spectrum was recorded with 5 μM of PviAPX at room temperature. The spectrum was normalized to have OD_404_ equal to one. The spectrum was collected in the range of 260 to 600 nm.

**Figure 2 ijms-24-01778-f002:**
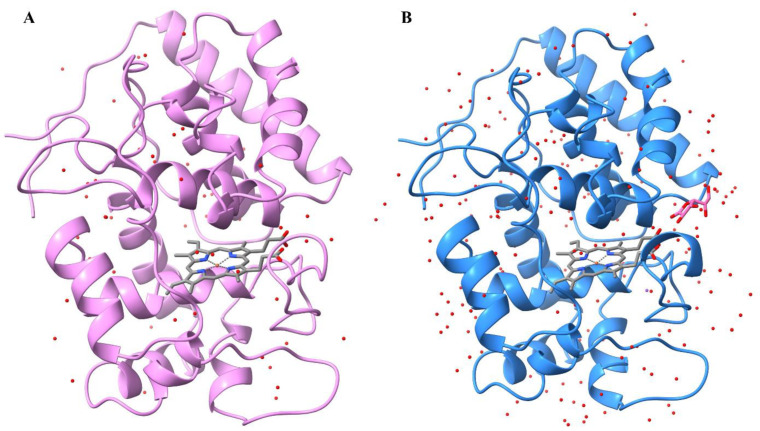
Overall structure of PviAPX. (**A**) PviAPX apo-form structure is shown in pink and heme is shown in gray. (**B**) PviAPX ascorbate complex structure is shown in blue, and ascorbate molecule is shown in the γ position. Red dots indicate water molecules. This figure was produced using the Chimera package Version 1.3 (UCSF, NIH P41 RR-01081).

**Figure 3 ijms-24-01778-f003:**
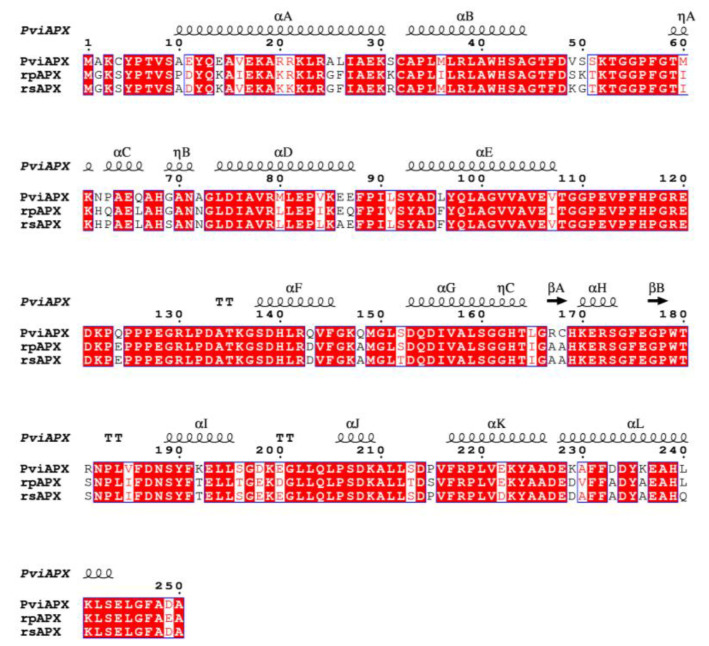
Sequence alignment of ascorbate peroxidases in monocots and dicots. Switchgrass APX (wild-type) was aligned with pea cytosolic APX (rpAPX) and soybean cytosolic APX (rsAPX). The secondary structure numbering followed the convention of peroxidase structure [40]. α refers to α-helices, β to β-strands, π to π-helices and η to 3_10_-helices. The conserved areas are enclosed in a blue frame. Red shaded, red characters and blue frames indicate descending similarity scores calculated with ESPript 3.0. This figure was generated using ESPript 3.0 (https://espript.ibcp.fr/ESPript/cgi-bin/ESPript.cgi accessed on 20 November 2022) [41].

**Figure 4 ijms-24-01778-f004:**
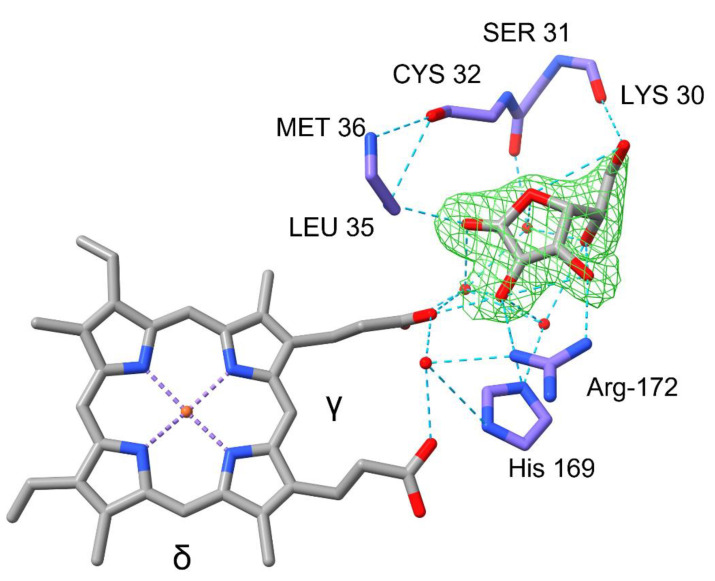
Ascorbate complex structure of PviAPX. The ascorbate molecule was shown in gray at γ-edge. This pocket was surrounded by Cys-32, Pro-34, His-169 and Arg-172. Nitrogen is shown in blue, oxygen in red and purple indicates carbon of residues. Purple dashed lines indicate metal coordination and blue dashed line indicate hydrogen bond network. This figure was produced using the Chimera package (UCSF, NIH P41 RR-01081).

**Figure 5 ijms-24-01778-f005:**
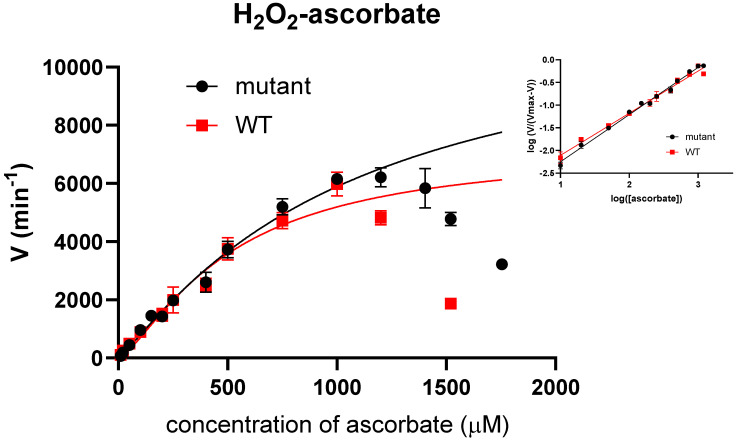
Kinetic assay of H_2_O_2_−dependent oxidation of ascorbate. Kinetic assay of PviAPX with ascorbate was performed with 1 mM H_2_O_2_ in 50 mM potassium phosphate buffer, pH 7.0. The concentration of ascorbate varied from 10 μM to 1800 μM, and the final concentration of PviAPX was 15 nM. The fitting was conducted with the ascorbate concentration ranging from 0 to 1200 μM. All the error bars indicate the standard deviation (*n* = 3). The insert shows the Hill plot fitted with linear regression.

**Figure 6 ijms-24-01778-f006:**
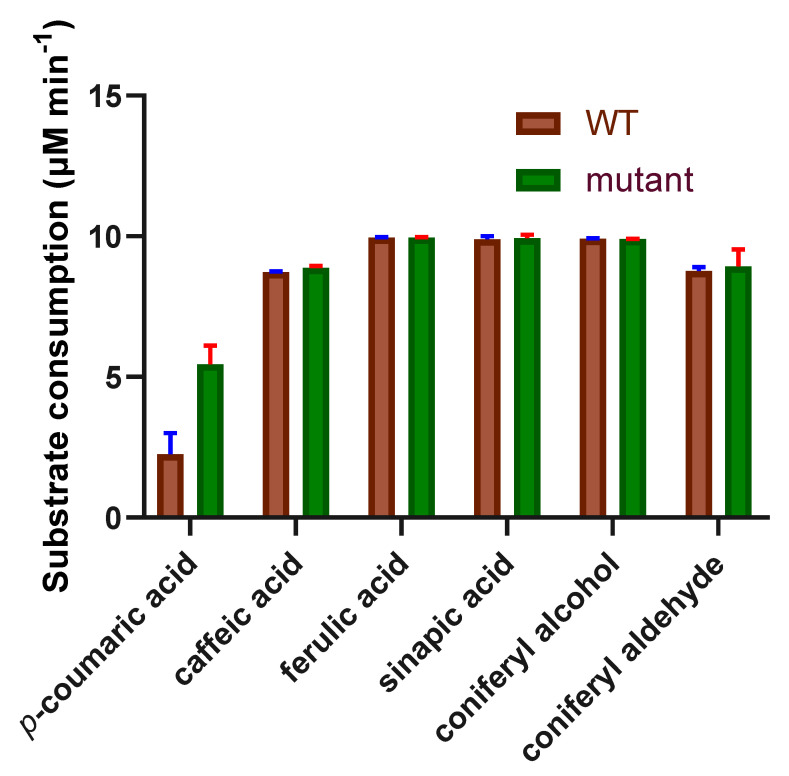
PviAPX kinetic assay of H_2_O_2_dependent oxidation of phenylpropanoids. The bar graph shows the H_2_O_2_-dependent oxidation of phenylpropanoids; consumption of *p*-coumaric acid, caffeic acid, ferulic acid, coniferyl alcohol and coniferyl aldehyde per minute were plotted with the standard error shown.

**Figure 7 ijms-24-01778-f007:**
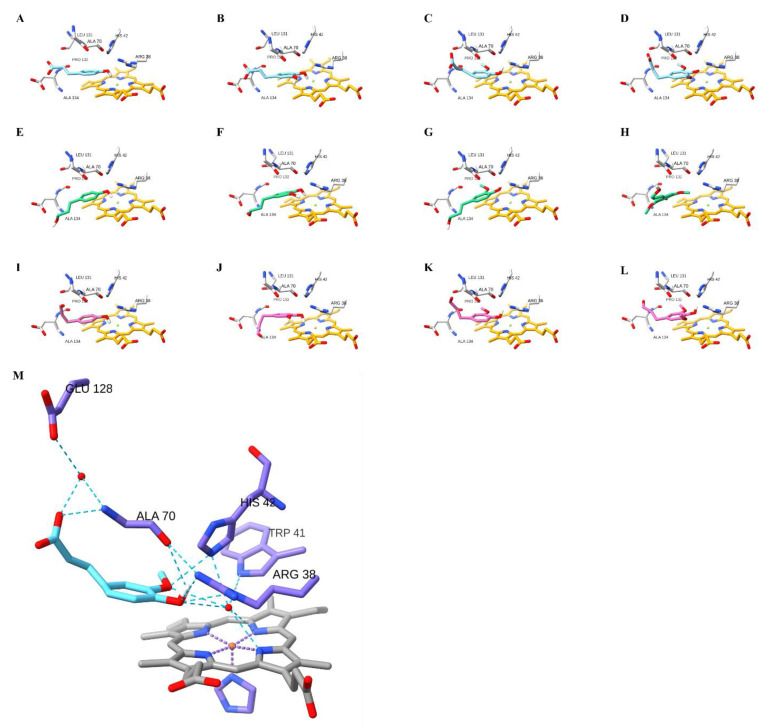
Molecular docking of phenylpropanoids in PviAPX. The acids (**A**) *p*-coumarate, (**B**) caffeate, (**C**) ferulate, and (**D**) sinapate are shown in light blue; (**E**) *p*-coumaryl alcohol, (**F**) caffeyl alcohol, (**G**) coniferyl alcohol, and (**H**) sinapyl alcohol are shown in green; aldehydes: (**I**) *p*-coumaryl aldehyde, (**J**) caffeyl aldehyde, (**K**) coniferyl aldehyde, and (**L**) sinapyl aldehyde are shown in pink. (**M**) shows the hydrogen bond network of the docked ferulate (light blue). Red color indicates oxygen and blue indicates nitrogen. This figure was produced using the Chimera package version 1.3 (UCSF, NIH P41 RR-01081).

**Table 1 ijms-24-01778-t001:** Crystallography statistics of PviAPX.

	Apo 8FF6	Ascorbate Complex 8FF7
Wavelength (Å)	1.000	1.000
Resolution range	49.23–2.193 (2.273–2.194)	49.36–2.494 (2.583–2.494)
Space group	P 1	P 1
Unit cell	78.267, 80.176, 80.166 104.552, 101.969, 110.727	78.558, 79.984, 80.147 104.327, 111.174, 101.782
Unique reflections	83,162 (7756)	56,445 (4919)
Multiplicity	2.5 (2.3)	3.3 (3.1)
Completeness (%)	97.17 (90.94)	96.73 (84.05))
Mean I/sigma(I)	6.45 (1.54)	8.06 (1.82)
Wilson B-factor	28.34	43.51
R-merge	0.1267 (0.5482)	0.1139 (0.4706)
CC1/2	0.98 (0.491)	0.985 (0.802)
R-work	0.1880 (0.2545)	0.1889 (0.2529)
R-free	0.2432 (0.3224)	0.2596 (0.3238)
Number of atoms		
Macromolecules and ligands	12,785	12,084
Solvent	911	282
RMS (bonds)	0.008	0.010
RMS (angles)	0.99	1.04
Ramachandran favored (%)	97.65	95.70
Ramachandran outliers (%)	0.00	0.07
Clash score	12.77	17.26
Average B-factor	33.03	49.58
Solvent	35.33	44.24

**Table 2 ijms-24-01778-t002:** Molecular docking energy of phenylpropanoid intermediates of the monolignol pathway.

Substrate	∆G Binding (kcal mol^−1^)
*p*-Coumarate	−5.5
*p*-Coumaryl alcohol	−5.5
*p*-Coumaryl aldehyde	−5.3
Caffeate	−5.8
Caffeyl alcohol	−5.7
Caffeyl aldehyde	−5.4 *
Sinapate	−5.8
Sinapyl alcohol	−5.6
Sinapyl aldehyde	−5.6
Ferulate	−5.7
Coniferyl alcohol	−5.7
Coniferyl aldehyde	−5.7

All the energy values are for the result with phenolic rings facing the heme group. * Indicates the energy is the second lowest energy among all docked positions.

## Data Availability

Sequence data from this publication can be found in the EMBL/GenBank data libraries under accession number XP_002468053.1. The structure discussed in this manuscript can be found at www.rcPv.org deposited under the corresponding PDB IDs: 8FF6 and 8FF7.

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
