# Peer review of "Activity of Cytosolic Ascorbate Peroxidase (APX) from Panicum virgatum against Ascorbate and Phenylpropanoids"

_ijms, 2023, doi:10.3390/ijms24021778_

Round 1

Reviewer 1 Report

Reviewer’s comments

The manuscript entitled “Activity of cytosolic ascorbate peroxidase (APX) from Panicum virgatum against ascorbate and phenylpropanoids” reports the crystal structures and enzymatic properties of APX from P. virgatum. The concerning issue described below should be clarify before acceptance.

<Major issue>

The experimental details for the results indicated in Figure 7 seems missing and should be described. If the section “3.3 Determination of oxidation products of PviAPX” corresponds to them, the substrates indicated are mismatched, and the concentration of the PviAPX used in the measurements of ferulic acid, sinapic acid and coniferyl alcohol was too high because all the substrates in the mixture were consumed within 10 min. In this case the rates are underestimated.

<Minor points>

Introduction: styles of the fonts (superscript, subscript and Italics) should be checked again.

Line 31: “equilibria”.

Line 50: select either “to” or “as”.

Line 109: “P. virgatum”.

Line 133-134: is it true that all the residues from the N-terminus Met to the C-terminus Ala were assigned against the electron density map?

Line 172-179: to clarify the explanation of this paragraph, a figure representing a close up view of the residues and electron density of the meatal ion described in this paragraph should be added as a supplemental figure.

Line 218-221: regarding the representation of the values of the kinetic constants, significant digit should be taken into account.

Line 232: add “mixture” after “reaction”.

Figure 7: change the color of the bar of either WT or mutant.

Line 259: correct “from with”.

Line 353: I could not find the results utilizing “tyrosine”.

Line 407-408: this sentence should be modified or removed.

Line 430-437: these sentences should be removed.

Figure A1: this figure should be moved to the supplemental data section.

Reviewer 2 Report

1.      Authors should please write the molecular formulas appropriately!!!. For example, check line 11 for hydrogen peroxide in the abstract section. Effect the corrections throughout the manuscript.

2.      The authors need to provide references for some important statements in the introduction section. For example, see Lines 52-54 etc.

3.      The methods require some further information, especially the sample size, equipment/instrument for the collection of sediment and leaves, how it was collected etc. Authors need to bring out or state detailed information on the design of this research.

4.      Please authors should provide some information on Panicum virgatum in the introduction section.

5.      Please the first mention of Escherichia coli in line 112 should be in full, and then the generic name can be abbreviated in the following sections

6.      Figure 2 in line 118 should be Fig. 1. Please authors should reconcile all the titles of their figures. Also, the titles should be summarized. For example, the explanation in Fig 2 after the first full stop should go to either the contents of the methodology or the results.

7.      I will advise authors should include the time and time frame for when this research was carried out. This should be captured in the material and methods section.

8.      Is there no data analysis in this study? Authors should specify that in their materials and methods under a section that should be titled “Data Analysis”

9.      Please provide a clearer image for Fig. AI.

10.  Authors should make sure all the generic names are italicized. For example, see line 71. Please correct this throughout the manuscript.

Reviewer 3 Report

The current article entitled “Activity of cytosolic ascorbate peroxidase (APX) from Panicum virgatum against ascorbate and phenylpropanoids” By Zhang et al., addresses the Optimization of lignin levels in biomass of plant cells. Which will help to mitigate the negative impact of lignin in the biochemical conversion process and will help increase the energy content of herbaceous biomass. As there are formatting and other mistakes so, the author must make suitable corrections.

 Abstract:

Line 13: Botanical name should be italics.

Line 117: PviAPX has the ability to oxidize a broad range of phenylpropanoids…

The phrase “has the ability” may be unnecessary and wordy. Consider replacing the phrase with “can”.

 Introduction:

Formatting mistakes for example: H2O2, ·O2−, should be H2O2 and O2- (uniform in all manuscript).

Line 33, 34, 43: Grammatical mistakes such as process, reaction, and follow. consider using plural nouns.

Line 52: “It is generally accepted that APX produces two monodehydroascorbate (MDHA) radials per one H2O2” correct the spelling of radicals.

Please write the scientific names in italics form. Line 88, 103 (Panicum virgatum)

Sentences are too long to understand in the overall manuscript. Please revised and rewrite.

The author should add the objectives of the study. Why the current study has been performed.

 Materials and methods:

The author did not explain the extraction of PviAPX.

Did the plant (Panicum virgatum) was collected or grown in the laboratory for the experiment?

LC-MS of phenylpropanoids: This portion of the methodology is confusing. Please revised it, because one cannot understand the sample preparation method of phenylpropanoids or their extraction from plants.

Results and Discussion:

Line 243: If the activity of both wild type and mutant Pvi APX is the same. Then what is the significance of this research?

As the binding energy of phenylpropanoids is too low can we consider this a good interaction?

The discussion section looks very shallow please improve this section.

I do not believe the article had a good scientific design, due to the insufficient conclusive power of the proposed approaches.

The author should extensively revise the results and discussion section and rewrite it.

Round 2

Reviewer 1 Report

A revised version of the manuscript entitled “Activity of cytosolic ascorbate peroxidase (APX) from Panicum virgatum against ascorbate and phenoylpropanoids“ requires revisions as pointed out below.

<Comments>

Line 35: insert a space between “2” and “M”.

Line 228-231: round the digit of the constants. For example, “12000 ± 4000 min-1”.

Line 228-231, Figure 6: the fitted curves indicate that the Vmax value of WT is apparently smaller than the mutant in Figure 6. But the kinetic values in the text are opposite. Recheck the values are correct or not.

Line 324: the style of “[56]” should be normal.

Reviewer 3 Report

The authors sufficiently revised the manuscript and now it is suitable to be accepted for publication in IJMS.

Author Response

Thank you for your critiques on the manuscript.